

**The pH dependency of the boron isotopic composition of diatom opal**
**(*Thalassiosira weissflogii*)**
Hannah K. Donald[1], Gavin L. Foster[1,*], Nico Fröhberg[1], George E. A. Swann[2], Alex J. Poulton[3,4], C. Mark
Moore[1] and Matthew P. Humphreys[1,5]
[1]School of Ocean and Earth Science, National Oceanography Centre Southampton, University of
Southampton, Southampton, SO14 3ZH
[2]School of Geography, University of Nottingham, University Park, Nottingham, NG7 2RD
[3]Ocean Biogeochemistry and Ecosystems, National Oceanography Centre, Southampton, SO14 3ZH
[4]The Lyell Centre, , Heriot-Watt University, Edinburgh, EH14 4AS
[5]School of Environmental Sciences, University of East Anglia, Norwich, NR4 7TJ
*Corresponding Author
**Abstract**
The high latitude oceans are key areas of carbon and heat exchange between the atmosphere and the
ocean. As such, they are a focus of both modern oceanographic and palaeoclimate research. However,
most palaeoclimate proxies that could provide a long-term perspective are based on calcareous
organisms, such as foraminifera, that are scarce or entirely absent in deep-sea sediments south of 50°
latitude in the Southern Ocean and north of 40° in the North Pacific. As a result, proxies need to be
developed  for the opal-based organisms (*e.g.* diatoms) that are found at these high latitudes, and
which dominate the biogenic sediments that are recovered from these regions. Here we present a
method for the analysis of the boron (B) content and isotopic composition ($\delta^{11}$B) of diatom opal. We
also apply it for the first time to evaluate the relationship between seawater pH  and $\delta^{11}$B and B
concentration ([B]) in the frustules of the diatom *Thalassiosira weissflogii,* cultured at a range of
$p$CO$_2$/pH. In agreement with existing data, we find that the [B] of the cultured diatom frustules
increases with increasing pH (Mejia et al., 2013). $\delta^{11}$B shows a relatively well-defined negative trend
with increasing pH; a completely distinct relationship from any other biomineral previously measured.
This relationship not only has implications for the magnitude of the isotopic fractionation that occurs
during boron incorporation into opal, but also allows us to explore the potential of the boron-based
proxies for palaeo-pH and palaeo-CO$_2$ reconstruction in high latitude marine sediments that have, up
until now, eluded study due to the lack of suitable carbonate material.




## 1. Introduction

The high latitude regions, such as the Southern Ocean and the subarctic North Pacific, exert key controls on atmospheric $CO_2$. Both areas are where upwelling of deep carbon- and nutrient-rich water occurs, which promotes outgassing of previously stored carbon to the atmosphere and nutrient fertilisation of primary productivity, in turn drawing down $CO_2$. The balance of processes involved in determining whether these oceanic regions are a source or sink of $CO_2$ are poorly understood, to the extent that the oceanic controls on glacial-interglacial pH and $p$CO$_2$ changes remain a subject of vigorous debate (*e.g.* Martin, 1990; Sigman and Boyle, 2000). Recently, several studies have shown how the boron isotope pH proxy applied to calcitic foraminifera successfully tracks surface water $CO_2$ content, and thus documents changes in air-sea $CO_2$ flux along the margins of these regions (*e.g.* Martínez-Botí et al., 2015; Gray et al. 2018). However, the lack of preserved marine carbonates in areas that are thought to be key in terms of glacial-interglacial $CO_2$ change (*e.g.* the polar Antarctic zone; Sigman et al., 2010) represents a currently insurmountable problem, and prevents the determination of air-sea $CO_2$ flux using boron-based proxies in regions that are likely to play the most important role in glacial-interglacial $CO_2$ change. There is therefore a clear need for the boron isotope palaeo-pH proxy to be developed in biogenic silica (diatom frustules, radiolarian shells), which is preserved in high-latitude settings, to better understand these key regions and their role in natural climate change.

The boron isotopic system has been used extensively in marine carbonates for the reconstruction of past ocean pH, and past atmospheric $CO_2$ (*e.g.* Hemming and Hanson, 1992; Pearson and Palmer, 2000; Hönisch and Hemming, 2005; Foster, 2008; Henehan et al., 2013; Chalk et al. 2017; Sosdian et al. 2018). Comprehensive calibration work has been completed for numerous species of foraminifera that are currently used in palaeoceanographic reconstruction (*e.g.* Henehan et al. 2016; Rae et al. 2011), and it has been shown that while $\delta^{11}B$ compositions are fairly similar among carbonates, species-specific differences exist in the relationship between the boron isotopic composition of dissolved borate and the $\delta^{11}B$ of foraminifera. Once this relationship is known, this $\delta^{11}B$-pH calibration can be applied to fossils found in deep-sea sediment cores, reliably reconstructing past ocean pH and $p$CO$_2$ (*e.g.* Hönisch and Hemming, 2005; Foster, 2008, Hönisch et al., 2009; Chalk et al., 2017). However, thus far the boron isotopic composition (expressed as $\delta^{11}B$) and B concentration ([B]) of the siliceous fraction of deep sea sediments remains poorly studied.

Early exploratory work by Ishikawa and Nakamura (1993) showed that biogenic silica and diatom ooze collected from modern deep sea sediments in the North and Equatorial Pacific had relatively high



boron contents (70-80 ppm), but a very light isotope ratio. For example, a diatom ooze was shown to
have a $\delta^{11}$B of -1.1 ‰ whilst radiolarian shells had a $\delta^{11}$B of +4.5 ‰. While some of this light $\delta^{11}$B may
have partly arisen due to clay contamination (reducing the diatom ooze sample by up to 3 ‰; Ishikawa
and Nakamura, 1993) it also likely reflects an opal:seawater isotopic fractionation arising from the
substitution of borate for silicate in tetrahedral sites in the opal (Ishikawa and Nakamura, 1993). A
similarly light $\delta^{11}$B was also observed in marine cherts from deep sea sediments by Kolodny and
Chaussidon (2004; -9.3 to +8 ‰), but these are unlikely to be primary seawater precipitates. A recent
culture study of the diatoms *Thalassiosira weissflogii* and *T. pseudonana* showed that the boron
content of cultured opal was significantly lower than suggested by the bulk sampling of Ishikawa and
Nakamura (1993) at around 5-10 ppm, increasing as pH increased from 7.6 to 8.7 (Mejia et al. 2013;
Supplementary Figure S1). This suggests seawater tetrahydroxyborate anion (borate; $B(OH)_4^-$) is
predominantly incorporated into the diatom frustule rather than boric acid ($B(OH)_3$), and implies there
is potential for the boron content of diatom opal to trace pH in the past (Mejia et al. 2013).

Here, the relationship between $\delta^{11}$B of the frustules of the diatom *T. weissflogii* and seawater pH is
investigated for the first time using a batch culturing technique and different air-CO$_2$ mixtures to
explore a range of pH (8.54 ± 0.57 to 7.48 ± 0.06). The aim of this study was also to develop a
methodology for measuring the boron isotopic composition of biogenic silica by MC-ICP-MS and apply
this method to explore the response of the boron based proxies ([B] and $\delta^{11}$B) in diatom frustules to
changing pH. Ultimately, we show how boron isotopes measured in diatom frustules may provide
further insight into boron uptake and physiological activity within diatoms, and we test the potential
of $\delta^{11}$B and boron content in diatoms as proxies for the ocean carbonate system.

**2.  Methods**
**2.1 Experimental Set up**
The centric diatom *T. weissflogii* (Grunow in van Heurck, PCC 541, CCAP 1085/1; Hasle and Fryxell,
1977) was grown in triplicate in K/1 enriched sterile and filtered seawater (0.2 $\mu$m; seawater sourced
from Labrador Sea; Keller et al., 1987) in 3 L glass Erlenmeyer flasks for a maximum of one week for
each experiment. Initial nutrient concentrations within the seawater before enrichment were
assessed on a SEAL Analytical QuAAtro analyser with a UV/vis spectrometer and ranged from 23.3 to
27.5 μM for nitrate(+nitrite), 4.3 to 5.4 μM for silicic acid, and 1.4 to 1.6 μM for phosphate. The culture
experiments were bubbled with air-CO$_2$ mixtures in different concentrations (sourced from BOC) to
provide a pH range at constant bubble rates, and every flask was agitated by hand twice daily to limit
algal settling and aggregation. The monocultures were grown in nutrient replete conditions at



constant temperature (20°C) and on a 12h:12 h light:dark cycle (with 192 $\mu$E m$^{-2}$ s$^{-1}$, or 8.3 E m$^{-2}$ d$^{-1}$
during the photoperiod). The diatoms were acclimated to each $p$CO$_2$ treatment for at least 10
generations before inoculating the culture experiment flasks. All culture handling was completed
within a laminar flow hood to ensure sterility. The flow hood surfaces were cleaned with 90% ethanol
before and after handling, as well as the outer surface of all autoclaved labware entering the laminar
flow hood such as bottles and pipettes.

The cultured diatom samples were collected by centrifugation at 96 h, during the exponential phase.
Each flask was simultaneously disconnected from the gas supply, and the culture was immediately
centrifuged at 3700 rpm for 30 minutes into a pellet, rinsed with MilliQ, and frozen at -20°C in sterile
plastic 50 mL centrifuge tubes. Around 10 mg of diatom was harvested in each experiment.

**2.2. Growth rate and cell size**
A 5 mL sub-sample was taken from each culture flask through sterilised Nalgene tubing into sterile
syringes, and sealed in sterile 15 mL centrifuge tubes. Triplicate cell counts using a Coulter
Multizier$^{TM3}$ (Beckman Coulter) were performed daily on each experimental flask. Growth rates were
calculated using equation 1:
$$\mu = (lnN_t - lnN_i)/(t - t_i) \qquad (1)$$
Where $N_i$ is the initial cell density at the start of the experiment ($t_i$) and $N_t$ is the cell density at time t.
Triplicate estimates of cell size were also determined using the Coulter Multizier$^{TM3}$, to determine
the mean cell size over time in each flask. Figure 1 shows that although there is no statistically
significant relationship between pH and diatom growth rate, cell size does show a small, but
statistically significant, positive slope.
**2.3 pH, DIC and δ$^{11}$B of the culture media**
A pH meter (Orion 410A) calibrated using standard National Bureau of Standards (NBS) buffers prior
to sample extraction was used to monitor the evolution of pH through the experiment on a daily
basis. For fully quantitative constraints on the carbonate system of the culture media, dissolved
inorganic carbon; DIC) was measured in triplicate, every other day, for each pH treatment (*i.e.* one
per experiment flask). The 100 mL bottles were filled to overflowing and immediately closed with
ground glass stoppers, then uncapped to be poisoned with 1 mL mercuric chloride (HgCl$_2$) to prevent
any further biologically-induced changes in DIC and stored sealed with Apiezon L grease in complete



darkness until analysis. Analysis of DIC was performed by acidification with excess 10% phosphoric
acid and $CO_2$ transfer in a nitrogen gas stream to an infrared detector using a DIC Analyzer AS-C3
(Apollo SciTech, DE, USA) at the University of Southampton. The DIC results were calibrated using
measurements of batch 151 certified reference material obtained from A. G. Dickson (Scripps
Institution of Oceanography, CA, USA). The accuracy of the DIC analysis was c. 3 µmol kg$^{-1}$.
Carbonate system parameters, including seawater $pCO_2$, were calculated using measured $pH_{NBS}$ and
DIC values, temperature, salinity and nutrients with the $CO_2SYS$ v1.1 programme (van Heuven et al.,
2011; using constants from Dickson, 1990; Lueker et al., 2000; Lee et al., 2010), which was also used
to convert pH from the NBS to the Total scale (used throughout).
All flasks were initially filled with media from the same large batch, and all culture treatments
therefore started with the same initial pH. The pH for all treatments was then altered by bubbling
through the different air-$CO_2$ mixtures, ranging from low pH (target = 1600 ppm, high $pCO_2$) to high
pH (target = 200 ppm, low $pCO_2$). Almost all treatments held relatively constant DIC and pH until the
final 24 hours of the experiment, when marked changes in DIC and pH in all culture treatments were
observed (Figure 2), which in most cases was likely due to the growth of diatoms and an associated
net removal of DIC, despite the constant addition of $pCO_2$. In order to account for these non-steady
state conditions of the carbonate system, the mean pH and $pCO_2$ of each treatment were calculated
based on the number of cells grown per 24 hours along with the pH/$pCO_2$ measured in that 24 hours,
thus adjusting for the observed exponential growth rate of *T. weissflogii* (Table 1).

The boron concentration of the culture media was not determined but is assumed to be the same as
Labrador seawater (~4.5 ppm; Lee et al. 2010). The boron isotopic composition of the culture media
was determined using standard approaches (Foster et al., 2010) to be 38.8 ± 0.19 ‰ (2 s.d.).
**2.4 Preparing cultured diatoms for $\delta^{11}$B and B/Si analysis**
In order to examine reproducibility and accuracy of our boron measurements, an in-house diatom
reference material was used to develop a method for measuring boron isotopes and boron
concentration in biogenic silica. A British Antarctic Survey core catcher sample (TC460) from core
TC460 in the Southern Ocean (-60.81534° N, -50.9851° E, water depth 2594 m) was used for this
purpose (supplied by C.-D. Hildebrand [British Antarctic Survey]). Although the diatom assemblage
was not characterised in the core catcher, the nearest sediment sample in the core is dominated by
*Hyalochaete Chaetoceros* resting spores, representing circa 70% of the total diatom content, with sea
ice and cool open water species making up the bulk of the remaining 30% (*e.g. Actinocyclus*
*actinochilus, Fragilariopsis curta, F. cylindrus, F. obliquecostata, Odontella weissflogii, Thalassiosira*



*antarctica*). A pure diatom sample of mixed species was separated from this bulk sediment and
cleaned of clay contamination at the University of Nottingham following an established diatom
separation technique (Swann et al., 2013). Briefly, the bulk sample underwent organic removal and
carbonate dissolution (using 30% $H_2O_2$ and 5% HCl), heavy liquid separation in several steps at
different specific gravities using sodium polytungstate (SPT), and visual monitoring throughout the
process to ensure the sample was free from non-diatom material, such as clay particulates. After the
final SPT separation, samples were rinsed thoroughly with MilliQ and sieved at 10 µm to remove all
SPT traces.

The culture samples and the diatom fraction from TC460 were first acidified ($H_2SO_4$), and organics
were oxidised using potassium permanganate and oxalic acid (following Horn et al., 2011 and Mejía
et al., 2013). The samples were rinsed thoroughly using MilliQ water via centrifugation and transferred
to acid-cleaned Teflon beakers. A secondary oxidation was completed under heat using perchloric
acid. Finally, the organic-free samples were rinsed thoroughly with MilliQ via filtration.

In the boron-free HEPA filtered clean laboratory at the University of Southampton, each sample was
dissolved completely in a gravimetrically known amount of NaOH (0.5 M from 10 M concentrated
stock supplied by Fluka) at 140°C for 6 to 12 h, and briefly centrifuged prior to boron separation to
ensure no insoluble particles were loaded onto the boron column. Anion exchange columns containing
Amberlite IRA 743 resin were then used to separate the matrix from the boron fraction of each sample
following Foster (2008). Briefly, the dissolved opal was loaded directly onto the column without
buffering and the matrix removed with 9 x 200 µL washes of MilliQ. This was collected for subsequent
analysis and the pure boron fraction was then eluted and collected in 550 µL of 0.5 M $HNO_3$ acid. The
level of potential contamination was frequently monitored using total procedural blanks (TPB)
measured in every batch of columns. The TPB comprised an equivalent volume of sodium hydroxide
(NaOH, 0.5 M) as used in the samples of each batch (ca. 0.2 - 4 mL). This was analysed following the
sample analysis protocols detailed below, and typically the TPBs for this work contained less than 40
pg of boron. This equates to a typical blank contribution of ca. 0.015%, which results in a negligible
correction and is therefore ignored here.

Prior to isotope analysis, all boron fractions were collected in pre-weighed acid cleaned Teflon beakers
and their mass was recorded using a Precisa balance. A 10 µL aliquot was taken and diluted with 490
µL 0.5 M $HNO_3$ in acid cleaned plastic centrifuge tubes (2 mL). This was then analysed using a Thermo
Fisher Scientific Element 2XR ICP-MS at the University of Southampton, with boron concentration





determined using standard approaches and a gravimetric standard containing boron, silicon, sodium,
and aluminium. In order to determine the B/Si ratio, and hence the B concentration of the opal, the
Si concentration must also be quantitatively measured. This is achieved here by using a known
concentration and mass of NaOH to dissolve each sample, and by measuring the Si/Na ratio the Si
concentration of each opal sample can be determined. From this, assuming a chemical formula of
$SiO_2.H_2O$ and a $H_2O$ content of 8% (Hendry and Anderson, 2013), the B content of the opal in ppm can
be estimated.  As detailed above, during the purification procedure, sample matrix was washed off
the column using MilliQ, and collected in pre-weighed acid cleaned Teflon beakers. These samples
were then diluted with 3 % $HNO_3$ enriched with Be, In and Re for the internal standardisation and
measured on the Thermo Scientific X-series ICP-MS. The standards run on the X-Series consisted of
varied concentrations of the gravimetric standard also used on the Element, containing B, Si, Na and
Al.

The boron isotopic composition of the biogenic silica samples was determined on a Thermo Scientific
Neptune MC-ICP-MS, also situated in a boron-free HEPA filtered laboratory at the University of
Southampton, following Foster (2008). Instrument induced fractionation of the $^{11}B/^{10}B$ ratio was
corrected using a sample-standard bracketing routine with NIST SRM 951, following Foster (2008).
This allows a direct determination of $\delta^{11}B$ without recourse to an absolute value for NIST SRM 951
(Foster, 2008) using the following equation, where $^{11}B/^{10}B_{standard}$ is the mean $^{11}B/^{10}B$ ratio of the
standards bracketing the sample of interest.

$$\delta^{11}B = \left[\left(\frac{^{11}B/^{10}B_{sample}}{^{11}B/^{10}B_{standard}}\right) - 1\right] \times 1000 \qquad (2)$$

The reported $\delta^{11}B$ is an average of the two analyses, with each representing a fully independent
measurement (*i.e.* the two measurements did not share blanks or bracketing standards). Machine
stability and accuracy was monitored throughout the analytical session using repeats of NIST SRM 951,
as well as boric acid reference materials AE120, AE121 and AE122 that gave $\delta^{11}B$ (± 2 s.d.) of -20.19 ±
0.20 ‰, 19.60 ± 0.28 ‰, and 39.31 ± 0.28 ‰, that are within error of the gravimetric values from Vogl
and Rosner (2012).

The reproducibility of the $\delta^{11}B$ and [B] measurements were assessed by repeat measurements of
TC460 of different total B concentration (11 to 34 ng of B). In order to assess the accuracy of this
method, we follow Tipper et al. (2008) and Ni et al. (2010) and use standard addition. To this end,
known amounts of NIST SRM 951 standard were mixed with known quantities of TC460. All mixtures



were passed through the entire separation and analytical procedure, including aliquots of pure
standard and sample. A sodium acetate - acetic acid buffer was added to all 951 boric acid used prior
to mixing, to ensure the pH was sufficiently elevated for the column separation procedure (following
Foster, 2008). The amount of biogenic silica matrix added to the columns for each mixture was kept
constant, so the volume added to the column was altered for each mixture accordingly. Uncertainty
in the $\delta^{11}$B calculated for each mixture was determined using a Monte Carlo procedure (n = 1000) in
R (R Core Team, 2019) propagating uncertainties, at 95% confidence, in known isotopes ratios (± 0.2
‰), sample concentration (± 6 %), and measured masses (± 0.5 %).

**3. Results and Discussion**
**3.1 Analytical Technique**
**3.1.1. Purification**
The Na, Si and Al concentration of the matrix fraction of several replicates of the diatom fraction of
TC460 are shown in Figure 3a-d.  Prior to purification, Na and Si concentrations were consistently
around 265 and 114 ppm respectively, whereas Al was more variable at 5-25 ppb. The boron content
of these matrix samples in all cases was at blank level.  The concentration of these elements in the
boron fraction is shown in Figure 3e-g, highlighting that the column procedure is sufficient to
concentrate boron and remove Na and Si which are both present at sub 5 ppb level (*i.e.* at less than
0.002 % of matrix concentration). The Al is likely present in the diatom frustule (*e.g.* Koning et al. 2007)
and is elevated in the boron fraction compared to the matrix fraction (Figure 3).  Diatom-bound Al is
likely present as the anion Al(OH)$_4^-$, hence its elevation in the boron fraction. Although this is a
detectable level of Al, it is unlikely that this level of contamination will influence the mass fractionation
of these samples when measured by MC-ICP-MS (Foster, 2008; Guerrot et al. 2010).

**3.1.2. Accuracy and Reproducibility**
Throughout the duration of this study, a single dissolution of the diatom fraction of TC460 was
measured 18 times in separate analyses at various concentrations, in order to assess external
reproducibility of this method. Carbonates generally have a reproducibility of ± 0.20 ‰ (2σ) at an
analyte concentration of 50 ppb boron using the MC-ICP-MS methods at Southampton (*e.g.* Chalk et
al. 2017). The repeated measurements of TC460 gave a reproducibility of ± 0.28 ‰ (2σ) over 18
samples, ranging from 19 ppb to 61 ppb (11 to 34 ng) boron (Figure 4). The similar $\delta^{11}$B regardless of
boron concentration analysed confirms that blank contamination during purification is not significant.
Figure 4 shows that there is also no correlation between Al content of the boron fraction and
measured $\delta^{11}$B, confirming Al contamination does not influence mass fractionation.



Figure 4 shows the results of the standard addition experiment, and when the uncertainty in the $\delta^{11}$B
of the mixture is considered, it is clear that nearly all the mixtures lie within error of the 1:1 line,
indicating that there is a lack of a significant matrix effect when analysing the $\delta^{11}$B of biogenic silica as
described herein. A least squares linear regression of the mixtures has a slope of 1.01 ± 0.07 and an
intercept of -0.15 ± 0.29 ‰, implying the approach is accurate to ± 0.29 ‰, which is remarkably similar
to the stated reproducibility of TC460 (± 0.28 ‰ at 2σ).

B and Si content were determined separately and combined post-analysis in order to estimate the
B/Si ratio for each sample and hence the B concentration. The reproducibility of this method was
tested using six repeats of the diatom fraction of TC460. The mean of all six measurements is 2.99 ±
0.64 ppm; (2σ; Figure 4), implying this multi-stage method of determining the B content of diatoms is
precise to ± 20 % at 95% confidence.

**3.2. Diatom Cultures**
**3.2.1.  Boron content of the frustule of *T. weissflogii***
The boron content of *T. weissflogii* increases as a function of pH from around ~1 ppm to ~4 ppm over
a range of average culture pH from 7.5 to 8.6 (Figure 5; Table 2). While this is lower by an order of
magnitude than the limited previous studies of boron in sedimentary diatoms (Ishikawa and
Nakamura, 1993), it is similar to boron concentration in the bulk diatom fraction of TC460 (Figure 5)
and to that observed in previous culturing studies of this diatom species (Figure 5; Meija et al. 2013).
In detail, however, our concentrations are around 2-3 times lower than Meija et al. (2013), perhaps
due to the different analytical methods used (laser ablation ICP-MS vs. solution here; Figure A1).
Despite the scatter between treatments (also seen in Meija et al. 2013; Figure A1), a least squares
regression through the treatments is significant at the 95% confidence level (y = 2.15x − 15.56, $R^2$ =
0.46, p = 0.015; Figure 5). The cause of this scatter between treatments is not known but a likely
contributor is the relatively high variability in the carbonate system which was observed in each
treatment due to the growth of the diatoms in this batch culture setup (Figure 2).

Boron is an essential nutrient for diatoms (Lewin, 1966) and it is likely that boric acid passively diffuses
across the cell wall to ensure the diatom cell has sufficient boron to meet its biological needs.
However, if boric acid were the sole source of boron for the diatoms measured here we might expect
a decrease in boron content as pH increases and external dissolved boric acid concentration declines
(Figure 6).




Several studies note that a number of higher plants have mechanisms for also actively taking up boron,
leading to large variations in internal boron concentrations (Pfeffer et al., 2001; Dordas and Brown,
2000; Brown et al., 2002). Indeed, on the basis of a similar dataset to that collected here, Meija et al.
(2013) suggested that borate is likely transported across the cell wall of *T. weissflogii* as some function
of external borate concentration, which shows a positive relationship with external pH (Figure 6). This
hypothesis is developed and discussed further in the next section.

**3.2.2.    Frustule $\delta^{11}B$ of *T. weissflogii***
The $\delta^{11}B$ of *T. weissflogii* are isotopically light compared to seawater (39.6 ‰; Foster et al., 2010), with
an average value across all treatments of -3.95 ‰ (Table 2). Despite the scatter between treatments,
similar to the [B] data, Figure 5 shows that there is a clear relationship between the $\delta^{11}B$ of the diatom
frustule and pH ($R^2$ = 0.43, p <0.01), albeit with a negative and relatively shallow slope (y = -2.37x +

315    15.34).


These results confirm that biogenic silica, free from clay contamination, has a very light boron isotopic
composition (Ishikawa and Nakamura, 1993). However, the observed relationship between $\delta^{11}B$ in *T.
weissflogii* and pH is radically different to that which is observed in carbonates (Figure 5), implying a
distinctive incorporation mechanism for boron into diatom opal. Much work has been carried out in
recent years to show that boron is incorporated in carbonates predominantly as the borate ion with
minor, if any, isotopic fractionation (*e.g.* see Branson, 2018 for a review). It is similarly thought that
the borate ion is incorporated into opal, in an analogous fashion to its incorporation into clays
(Ishikawa and Nakamura, 1993; Kolodny and Chaussidon, 2004). However, such a mechanism in
isolation would only be able to generate $\delta^{11}B$ in opal of ~13 ‰ (at lowest pH). Given the
preponderance of isotopically light diatoms, radiolaria and chert $\delta^{11}B$ in the literature (including this
study; Kolodony and Chaussidon, 2004; Ishikawa and Nakamura, 1993), it is therefore likely that there
is an additional light isotopic fractionation of boron on its incorporation into opal, although its
absolute magnitude is currently unknown (Kolodony and Chaussidon, 2004).

Without knowledge of the isotopic fractionation of boron on incorporation into biogenic silica, the
interpretation of our new $\delta^{11}B$ data is therefore challenging. This difficulty is further increased given
that the fluid in the silica deposition vesicle (SDV) in diatoms is unlikely to have the boron isotopic
composition of external seawater, and is likely at a relatively acidic pH (~5.5; Meija et al. 2013) to
promote polymerisation of $Si(OH)_4$. Nonetheless, the broad similarity between the $\delta^{11}B$ of our cultured



*T. weissflogii* with the bulk diatom fraction measured here from sample TC460, and the bulk diatom
fraction and radiolarian skeleton measured by Ishikawa and Nakamura (1993), suggests a large part
of the light isotopic composition of biogenic silica is driven by the isotopic fractionation on
incorporation rather than "vital effects" relating to the $\delta^{11}$B and pH of the SDV in the different species
and organisms. That being said, the >3‰ range between different pH treatments in *T. weissflogii* and
the >10 ‰ difference between our *Chaetoceros* dominated bulk diatom fraction from TC460 and the
cultured *T. weissflogii,* as well as the negative relationship between pH and diatom $\delta^{11}$B (Figure 5),
argues against a simple two-step model involving the incorporation of seawater borate ion and a fixed
isotopic fractionation on incorporation.

The $\delta^{11}$B of the fluid from which our *T. weissflogii* precipitated their frustules can be calculated if
we assume the pH in the SDV of our *T. weissflogii* is 5.5 across all our treatments (Mejia et al.,
2013). This suggests that the isotopic composition of this fluid is lighter than seawater, even if we
assume an arbitrary -10 ‰ isotopic fractionation on incorporation (Figure 7a). Furthermore, the
$\delta^{11}$B of the SDV fluid is inversely correlated with the $\delta^{11}$B of either dissolved borate or dissolved
boric acid (Figure 7a).
As discussed above, Mejia et al. (2013) suggested that there are two sources of boron in a diatom
cell: (i) passively diffused and isotopically heavy boric acid; and (ii) actively incorporated
isotopically light borate ion (see Figure 6). Assuming that: (a) no additional fractionation occurs
during uptake and diffusion; and (b) only the borate ion is incorporated into the frustule, we can
calculate the relative contribution of these two sources of boron as a function of external pH
(Figure 7b). This treatment shows that the relative concentration of borate derived boron in the
SDV fluid increases as external pH increases, though the absolute values here are a function of
the magnitude of the isotopic fractionation on incorporation, and so we only have confidence in
the trends shown in Figure 7b. Nonetheless, given that dissolved boric acid concentration
decreases and dissolved borate increases as pH is increased (Figure 6), this is perhaps not
surprising. This finding is also entirely compatible with the trend of increasing boron content of
*T. weissflogii* observed as pH increases (Figure 5).
Mejía et al. (2013) proposed that the enrichment of borate ion into the SDV of *T. weissflogii* and *T.*
*pseudonana* was the result of the active co-transport of borate ion with bicarbonate ion by
bicarbonate transporter proteins. Borate is transported because of its similar charge and size to $HCO_3^-$
and the phylogenetic similarity between bicarbonate and borate transporters (Mejía et al., 2013). In



this model, as external borate ion concentration increases, the borate leak into the diatom cell is also
increased. An additional factor is the $HCO_3^-$ transport, which may be proportionally up-regulated as
external $CO_2$ content decreases (as external pH increases) in order to provide the diatom cell with
sufficient carbon (Mejía et al., 2013). This may therefore offer an additional factor driving an elevation
of the borate content of the SDV as pH increases (Mejía et al., 2013). Regardless of the exact
mechanism, a SDV fluid that displays a boron isotopic composition as an inverse function of pH is
required to explain the observed $\delta^{11}B$ composition of the frustule of *T. weissflogii* measured here. A
simple model whereby external borate ion is an increasingly important contributor to the boron in the
SDV as pH increases is able to explain the observed dependency of boron content and $\delta^{11}B$ on pH.
However, a more complete model of the boron systematics in diatom opal requires a better
understanding of the isotopic fractionations on incorporation of boron into biogenic silica, the
environmental controls on this fractionation, and partitioning of the boron into biogenic silica.

**3.2.3.    Boron based pH proxies in diatom opal**
The $\delta^{11}B$-pH and B-pH relationships derived here for *T. weissflogii* potentially offer two independent
means to reconstruct the past pH of seawater, particularly in those regions key for $CO_2$ and heat
exchange where foraminifera are largely absent (e.g. the high Southern and Northern latitudes).
However, the current calibrations (Figure 5) are relatively uncertain and this may preclude their
application to some situations. For instance, recasting the $\delta^{11}B$-pH relationship in terms of $\delta^{11}B$ as the
dependent variable and using a regression method that accounts for uncertainty in X and Y variables
(SIMEX; Carroll et al., 1996) gives the calculated residual pH of the regression as ± 0.28 pH units. For
the [B] vs. pH relationship, this uncertainty is ± 0.36 pH units. At TA or DIC typically found in the surface
ocean such a variability in pH would translate to estimated seawater $pCO_2$ of ca. ± 250 ppm.  Although
encouraging, this treatment suggests that additional work is needed before the relationship between
$\delta^{11}B$ and boron content of diatom opal and seawater pH is a sufficiently precise proxy for a fully
quantitative past ocean pH. In particular, future culturing efforts should aim to more carefully control
the pH of the culture media. This could be achieved by either using larger volume dilute batch cultures,
and/or by harvesting the diatoms earlier in the experiment prior to any significant drift in the
carbonate system, or more robustly through using a steady state chemostat method (Leonardos and
Geider, 2005).

**4.  Conclusions**
In the first study of its kind, using a modified version of the carbonate boron purification technique of
Foster (2008), we show that the $\delta^{11}B$ of *T. weissflogii* opal is pH sensitive but isotopically light (-3.95





‰ on average), and has an inverse relationship with external seawater pH. Using a novel ICP-MS
method we also show that the boron content of *T. weissflogii* opal increased with increasing pH,
supporting the only other study investigating boron in diatoms (Mejía et al., 2013). This suggests that
borate is incorporated into the diatom frustule as the dissolved borate abundance increases with
external pH. A simple model is presented, based on Mejía et al. (2013), which implies both of these
findings could be due to the boron in the SDV having two distinct sources: external boric acid and
external borate ion, with the balance of each source changing with external pH. While these results
are encouraging and suggest that the boron proxies in diatom opal may hold considerable promise as
a tracer of past ocean pH, more work is needed to fully understand the boron systematics of diatom
opal. In particular, there is an urgent need to place boron in opal on a firmer grounding with
precipitation experiments in the laboratory at controlled pH to determine the magnitude of boron
isotopic fractionation on boron incorporation into opal, and subsequently the dependence of this
fractionation on other environmental factors.

**Acknowledgements**
We wish to thank Claus-Dieter Hildebrand for suppling the diatom rich sediment sample TC460.  John
Gittins, Mark Stinchcombe, Chris Daniels and Lucie Munns are acknowledged for their help during the
culturing and subsequent nutrient and carbonate system analysis.  Heather Stoll is also thanked for
her useful discussions on this topic. Financial support for this study was provided by the Natural
Environmental Research Council (UK) to H.K.D. (grant number 1362080) and to G.L.F. (NE/J021075/1).













**Figure Captions**

**Figure 1**. Diatom growth rate and cell size as a function of pH labelled according to $CO_2$ treatment. Linear least squares regressions, including $R^2$ and p-values are also shown.

**Figure 2**: Each culture treatment labelled according to target $pCO_2$ and showing the evolution in the culture media through the experiment. All treatments exhibit changes in DIC due to diatom growth balanced with the input of $pCO_2$. The higher $pCO_2$, the more DIC increases towards the end of the experiment.

**Figure 3**: (a-d) Concentration of Na, Si, Al and B in the Matrix Fraction by ICP-MS. These analyses suggest blank levels of B are present in the matrix washed off the Amberlite IRA 743 resin-based column. (e-f) Concentration of the Na, Si and Al in the boron fraction indicating blank levels of Na (ca. 1.7 ppb) and Si (ca. 1.9 ppb), and a higher concentration of Al (ca. 68 ppb) are present.

**Figure 4:** (A) The reproducibility of the TC460 diatom core catcher in-house standard. This shows all samples lie within error of the mean (5.98 ‰ ± 0.28 ‰, 2σ) at varied concentrations. This compares well to carbonates (2σ = 0.20 ‰). (B) Aluminium concentration of the B fraction from TC460 (as ppb of the solution analysed for $\delta^{11}B$) shows no correlation with $\delta^{11}B$, likely suggesting there is no significant effect on mass fractionation for this level of Al. (C) The results of the standard addition experiment. The blue line is a least squares regression between the measured $\delta^{11}B$ of each mixture (green circles) and the calculated $\delta^{11}B$ of that mixture given known end-member values (end members shown as blue circles). $R^2 = 0.97$, $p < 0.0001$, slope = 1.01 ± 0.07 and intercept = -0.15 ± 0.29. 1:1 line is shown as a black line and dotted blue lines show the 95% confidence limit of the regression. Note that the end members were not used in the regression. (D) B content in ppm of six repeat samples of the diatom fraction of TC460. The black line indicates the mean value, and the grey lines show 2σ, of 2.99 ± 0.64 ppm.

**Figure 5**: (A) $\delta^{11}B$ of *T. weissflogii* diatom opal plotted against aqueous borate, labelled according to $pCO_2$ treatment. Also shown are published deep sea coral *Desmophyllum dianthus* (Anagnostou et al., 2012) and foraminifera $\delta^{11}B$ (*Globigerinoides ruber* and *Orbulina universa*; Henehan et al., 2013; Henehan et al., 2016, respectively). Least squares regression lines are also shown. (B) Boron content of cultured *T. weissflogii* diatom opal as a function of pH labelled according to $pCO_2$. A least squares regression with 95% confidence interval is also shown. (C) *T. weissflogii* opal $\delta^{11}B$ against pH of each treatment demonstrating a statistically significant negative relationship. Uncertainty in all points is shown at the 95% confidence level. In some cases, the error bars are smaller than the symbols.

**Figure 6**: Plots describing (A) the pH-dependent relationship between the abundance of aqueous boron species, and (B) the isotopic fractionation observed between boric acid ($B(OH)_3$; red) and borate ($B(OH)_4^-$; blue) at T = 25 °C and S = 35.



**Figure 7**: (A) Back-calculated $\delta^{11}$B of the silica deposition vesicle (SDV), and (B) the fraction of boron
in the SDV that is derived from external borate. In (A) the diatom $\delta^{11}$B data are shown as grey circles
and the calculated $\delta^{11}$B of the SDV as blue circles. Included in this model is an arbitrary -10 ‰
fractionation between the $\delta^{11}$B of the SDV and the opal precipitated. The fraction of borate in the
SDV in (B) is a function of this assumption so these absolute values should be taken as illustrative
only.





**Tables**

| Treatment | $pCO_2$ (ppm) | $2\sigma$ | pH | $2\sigma$ | DIC ($\mu M$) | $2\sigma$ | $HCO_3^-$ ($\mu M$) | $2\sigma$ | Growth rate ($d^{-1}$) |
|---|---|---|---|---|---|---|---|---|---|
| 200 | 125 | 8 | 8.53 | 0.73 | 1925 | 61 | 1091 | 59 | 1.03 |
| 280 | 244 | 73 | 8.25 | 0.41 | 2165 | 113 | 1521 | 260 | 1.03 |
| 400 | 267 | 28 | 8.25 | 0.44 | 2400 | 115 | 1728 | 107 | 0.96 |
| 800 | 809 | 62 | 7.83 | 0.24 | 2525 | 56 | 2206 | 69 | 1.01 |
| 1600 | 2117 | 40 | 7.48 | 0.08 | 2791 | 21 | 2628 | 22 | 1.01 |

*Table 1: Mean carbonate system parameters experienced under the average growth*
*conditions as calculated for each culture treatment on the basis of the number of cells*
*grown in each 24-hour period of the batch experiment.*

| Treatment | pH (Total scale) | pH $2\sigma$ | $\delta^{11}B$ | $\delta^{11}B$ $2\sigma$ | $\delta^{11}B$ sw borate | [B] ppm |
|---|---|---|---|---|---|---|
| 200 | 8.55 | 0.63 | -5.51 | 0.21 | 24.20 | 3.15 |
| 200 | 8.54 | 0.62 | -5.40 | 0.21 | 24.00 | 2.81 |
| 280 | 8.27 | 0.35 | -5.05 | 0.20 | 20.00 | 3.72 |
| 280 | 8.18 | 0.25 | -5.66 | 0.21 | 18.80 | 0.93 |
| 280 | 8.30 | 0.42 | -5.79 | 0.21 | 20.50 | 1.04 |
| 400 | 8.26 | 0.38 | -3.64 | 0.20 | 19.90 | 3.37 |
| 400 | 8.24 | 0.36 | -3.57 | 0.21 | 19.60 | 1.26 |
| 400 | 8.25 | 0.36 | -2.41 | 0.21 | 19.70 | 2.68 |
| 800 | 7.85 | 0.22 | -2.93 | 0.19 | 15.40 | NA |
| 800 | 7.82 | 0.18 | -2.80 | 0.22 | 15.20 | 0.78 |
| 800 | 7.82 | 0.20 | -3.08 | 0.21 | 15.20 | 1.11 |
| 1600 | 7.48 | 0.06 | -1.94 | 0.20 | 13.30 | 0.74 |
| 1600 | 7.48 | 0.07 | -3.62 | 0.21 | 13.30 | 0.91 |

*Table 2. Treatment name and pH with $\delta^{11}B$ and [B] for cultured T. weissflogii.*








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





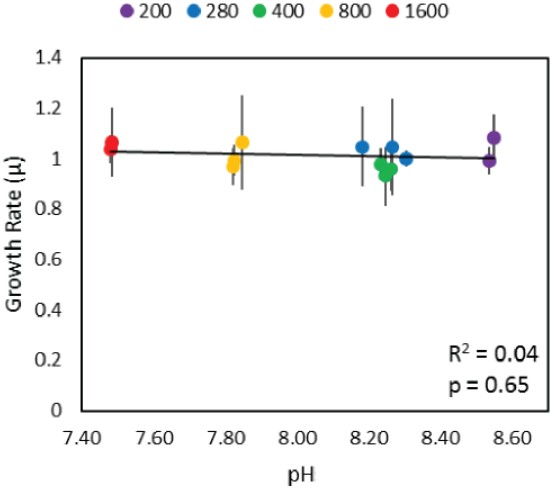
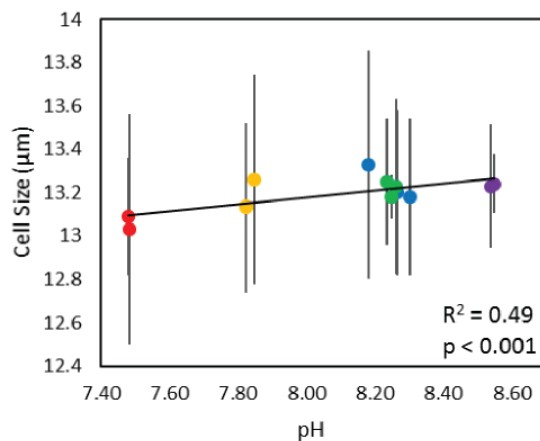

**Figure 1**





Figure 2





Figure 3.



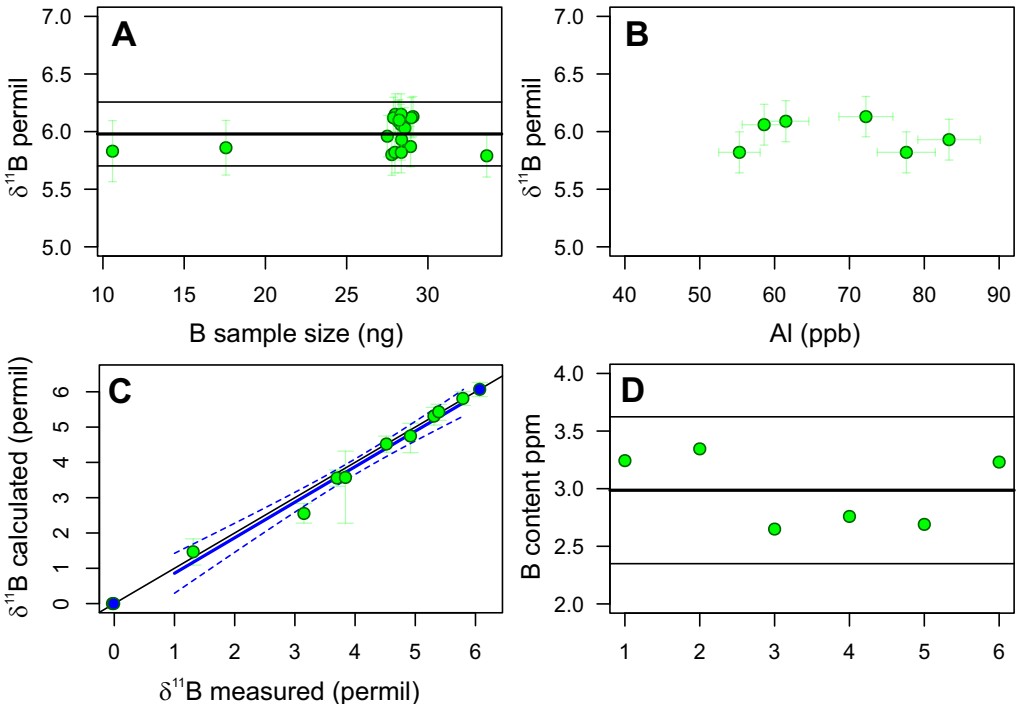

Figure 4



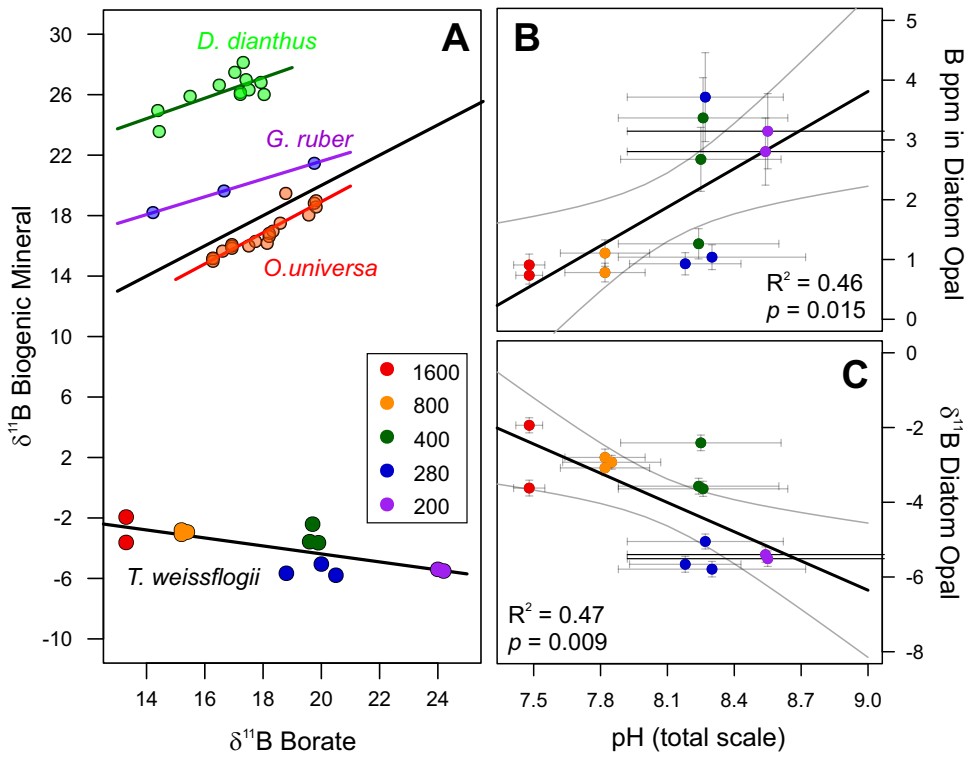

Figure 5.





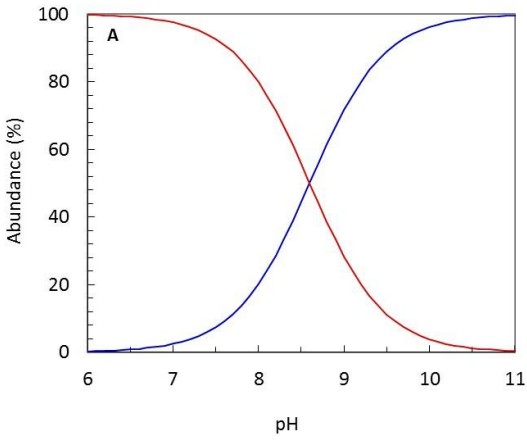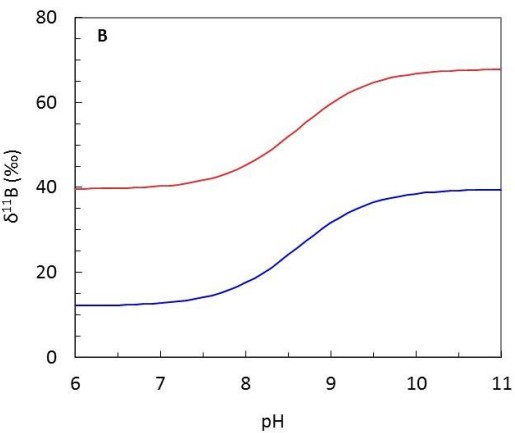

Figure 6.



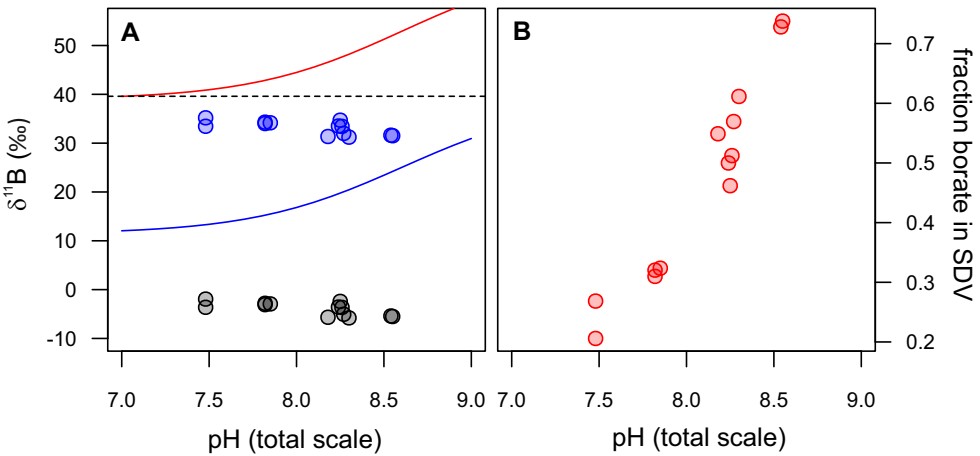

Figure 7.