# Peer review of "The pH dependency of the boron isotopic composition of diatom opal"

_Biogeosciences, 2019_

## Referee Comment (RC1) · Jan Fietzke (Referee) · 5 Sep 2019

This manuscript focusses on a truely challenging task. The goal of using biogenic opal, i.e. diatoms, as a seawater pH proxy archive is timely. I really applaude the author's efforts and think, this study has the potential to trigger further work into that particular application. Thus, I strongly believe the study is of interest for the readership ob BG and will be published after major revision.

My main problem at this stage is to evaluate the data and in particular the potential for analytical biases. I had the opportunity to analyze cultured T. weissflogii samples for their boron isotopic composition using LA-MC-ICPMS about 7 years ago. On av-

erage those analyses resulted in d11B of 14.0 +- 1.1 (1sd). Unfortunately, the results appeared too imprecise to be useful and never got published... Nevertheless, the significant difference of both, own data and those reported in this manuscript, is quite striking to me. Don't get me wrong, if we had been fully confident in our old data, we would have published... ;)

Thus, I've got to trust this new data as presented and need to evaluate the proposed interpretation. I would like to see in an additional figure a direct comparison of the [B] vs. pH systematic reported by Meija et al. (2013) and this study. The authors suggest the differences may be due to the use of LA and conventional ICPMS. I do not think, the LA results published by Meija et al. (2013) are inaccurate. Measuring [B] in silicates using silicate standards for normalization leaves little wiggle room for matrix effects. So, this would indicate three possibilties: a) samples for the older LA study had not been cleaned sufficiently (which I doubt strongly) b) the sample preparation used in this study resulted in a loss of boron or c) some details in the culturing setups resulted in this observable differences.

I would also be interested to see a figure displaying d11B vs. [B]. From figure 5 it appears there may be a stronger correlation of those two parameters than the ones of each of both vs. pH. The model proposed to explain the data (including the -10permill offset during incorporation into opal) is not really satisfying. Sorry, maybe I do not get it propperly. But I struggle with understanding how this model can produce d11B values which are some 15-18permill lighter than the lightest seawater-derrived borate d11B values (likewise for pH5.5). This would need a better, more detailed description, maybe including a schematic figure for a better conceptual understanding.

Overall, pretty interesting material and worth getting published. Perhaps starting very controversial future debates, if heavier diatom d11B data are reported by other groups in the future. But fine, we need to get this started.

---

## Referee Comment (RC2) · Joji Uchikawa (Referee) · 11 Dec 2019

1] General Comment (Overall quality of the discussion paper).

In this cutting-edge work, the authors explored the potential of boron isotopes ($\delta$11B) in diatoms to be employed as an indicator for seawater pH. The $\delta$11B paleo-pH technique has been extensively tested for foraminifers over the years. Yet, their tests are often scarce/absent in sediments from high latitude regions, where air-sea CO2 exchange is particularly dynamic. Given the abundance of diatoms in those areas, robust $\delta$11B calibrations against pH could eventually provide new insights into global carbon cycling, especially on glacial-interglacial timescales.

[Figure]

It is unfortunate that carbonate chemistry of the culture media underwent substantial drifts (by more than 0.5 pH units in some cases), which, as the authors admit, led to some uncertainties in relating diatom B contents and $\delta$11B to seawater pH. Thus, at this stage, the results from this study seem to leave more questions than the clear understanding on how B systematics works in diatoms. However, there is no question that this paper represents an important first step and it is highly worthy of publication.

The manuscript is generally well written. I raise some questions in this review (given below), but I must admit that they most likely stem from my limited familiarity with the process and mechanism of frustule formation in diatoms. Nonetheless, I think the readability of this paper will improve if they are further explained/clarified.

2] Specific comments (Individual scientific questions/issues).

- I would love to see figure-S1 in the main body of the paper (with proper data legend), which nicely compares the [B] vs pH relationship revealed here and previously in Mejía et al. (2013). Or alternatively, I recommend to include the data by Mejía et al. (2013) in Figure 5B.

- Figure 5: It might be worthwhile to add horizontal error bars (uncertainties for the projected $\delta$11B values for borate due to the drifts in pH). In addition, I am curious if the authors considered more of a threshold-type response (e.g., step-wise increase) for the relationship between diatom B contents and seawater pH (Panel B), rather than linear regression?

- Regarding the in-house TC460 standard used for validating the analytical methods. It was clear to me that the variations in B contents in the TC460 shown in Figure 4C were due to supplemental additions of different amounts of the NIST SRM951 reference material. But this appears not the case for the variations in B contents shown in Figure 4A. Then, do they simply reflect the differences in the amounts of TC460 dissolved?

- I feel the manuscript will significantly benefit from having a short paragraph describing

the process of how diatoms produce their frustules, including how silicic acid is gained from seawater into the cell and what happens afterwards (especially in the Silicate Deposition Vesicles: "SDV") to deposit silica frustules. Perhaps, the paragraph should also have brief description of carbon acquisition for photosynthesis (Line 364-371), to which an active intake of seawater borate is linked (Line 364-372).

- I have several questions and unclear points in the Section 3.2.2. (also Fig. 7), where possible scenarios for B incorporation in diatoms are discussed. The authors consider contributions of both B(OH)3 (via passive diffusion) and B(OH)4− (via active uptake) from seawater (Line 353-354). Also considered is a potential (arbitrary) 11B fractionation of −10‰ between B in SDV and B in frustules. But it was not clear to me how this fractionation was implemented in the calculations presented in Figure 7 (Line 467-468), which is showing nearly 40‰ offset between B in SDV and frustules (as opposed to 10‰ shown by blue vs grey circles). In Line 354-355, the authors provide two assumptions for the calculations presented in Figure 7, which are; (#1) no additional fractionation during active uptake and passive diffusion of B from seawater into the diatom cell and (#2) only B(OH)4− is incorporated into the frustules. I think it is safe for authors to acknowledge that assumption #1 is an entirely open question at this stage. Furthermore, I wonder how assumption #2 is possible. I presume SDV plays central roles for frustule formation (silica polymerization?). But it is mentioned that the internal pH of SDV should be around 5.5 (Line 334, Line 347). At such pH, essentially all of dissolved B should exist as B(OH)3, not B(OH)4− (Fig. 6). How can you incorporate something that does not exist?

3] Technical corrections.

Line 71: Comma after "(Ishikawa & Nakamura, 1993)".

Line 74: Could you be more specific on "seawater precipitates"?

Line 94: Could you provide the definition for "K/1 enriched" seawater?

Line 99: Could you provide the definition for "BOC"?

Line 109: exponential "growth" phase?

Line 251: Comma after "Si".

Line 287: "(Figure 5)". I think this should be Figure 4D, which shows B content in the in-house TC460 standard.

Line 353: "actively incorporated". I feel "transported" or something might be a better word choice over incorporated. Throughout the manuscript, the word "incorporated/incorporation" is used to refer to B in the frustules. But here, what is discussed is the very first step of B acquisition from seawater into the cell, for which I think a different wording is preferred to avoid confusion.

---

## Author Comment (AC1) · 12 Feb 2020

We thank the reviewer for his comments and in particular for his recognition that, while this manuscript is a start of the exploitation of boron isotopes in diatoms, it is not the final word on the matter. We respond to each of his comments in turn below.

RC1: "I had the opportunity to analyze cultured T. weissflogii samples for their boron isotopic composition using LA-MC-ICPMS about 7 years ago. On average those analyses resulted in d11B of 14.0 +- 1.1 (1sd). Unfortunately, the results appeared too imprecise to be useful and never got published... Nevertheless, the significant difference of both, own data and those reported in this manuscript, is quite striking to me."

This is indeed quite a difference, but without more detail it is hard for us to critically evaluate this observation. However, we would like to point out that: (i) we carried out an extensive cleaning protocol to remove residual organic material; (ii) we carried out an extensive investigation of our protocol including ensuring little to no boron was lost during the purification process; and (iii) our standard addition tests support the conclusion that our d11B analytical method is accurate. It is also perhaps worth noting that all published d11B measurements to date are also isotopically light, like our results – though we acknowledge that this is a rather limited dataset to make such comparisons. In the future we would welcome engaging with the community to further explore the analytical accuracy of d11B in opal-matrices by various analytical techniques.

RC1:"I would like to see in an additional figure a direct comparison of the [B] vs. pH systematic reported by Meija et al. (2013) and this study."

This was included in the original manuscript as a supplementary figure. Given this comment (and a similar one by reviewer RC2) we will bring this comparison into the main text.

RC1: "The authors suggest the differences may be due to the use of LA and conventional ICPMS. I do not think, the LA results published by Meija et al. (2013) are inaccurate."

This is not actually what we say in the manuscript, we were careful not to apportion cause and instead we said the following: "In detail, however, our concentrations are around 2-3 times lower than Meija et al. (2013), perhaps due to the different analytical methods used (laser ablation ICP-MS vs. solution here. . .".

RC1: "So, this would indicate three possibilties: a) samples for the older LA study had not been cleaned sufficiently (which I doubt strongly) b) the sample preparation used in this study resulted in a loss of boron or c) some details in the culturing setups resulted in this observable differences."

Since we have not done a direct comparison of methods for determining B/Si it is hard to determine the specific cause. However, given this comment, we will briefly discuss these possibilities in the manuscript, expanding on the observed discrepancy (in absolute B/Si but not in the relationship between B/Si and pH).

RC1:"I would also be interested to see a figure displaying d11B vs. [B]. From figure 5 it appears there may be a stronger correlation of those two parameters than the ones of each of both vs. pH."

We will include this in the revised manuscript.

RC1: "The model proposed to explain the data (including the -10permill offset during incorporation into opal) is not really satisfying. . . This would need a better, more detailled description,maybe including a schematic figure for a better conceptual understanding."

The incorporation of a schematic figure may work well in aiding the understanding of the model we propose. We are happy to include one in the revised manuscript.

---

## Author Comment (AC2) · 12 Feb 2020

We thank this reviewer for his comments and we are pleased that he favours publication and recognises the importance of this contribution. In order to harvest sufficient biomass from our dilute batch cultures we had to let them continue longer than we would have if we were going to avoid significant pH drift. We agree with the reviewer that this is a weakness of this current study but we do acknowledge this in the manuscript and recommend a different approach for subsequent studies. We respond to the rest of RC2's in turn below.

RC2:"I would love to see figure-S1 in the main body of the paper (with proper data

[Figure]

legend), which nicely compares the [B] vs pH relationship revealed here and previously in Mejía et al. (2013). Or ternatively, I recommend to include the data by Mejía et al. (2013) in Figure 5B"

As mentioned in our response to RC1, we will now include these data as a figure in the main body of our revised manuscript.

RC2: "Figure 5: It might be worthwhile to add horizontal error bars (uncertainties for the projected 11B values for borate due to the drifts in pH)."

This is a good idea and something we will do in the revised manuscript.

RC2: "In addition, I am curious if the authors considered more of a threshold-type response (e.g., step-wise increase) for the relationship between diatom B contents and seawater pH (Panel B), rather than linear regression?"

This is an interesting point but given the uncertainties in both pH and [B] we would prefer not to overly fit curves to the observed dataset. The statistics suggest a linear fit describes the data adequately (albeit with some scatter) and such a two stepped relationship is not consistent with the data from Mejia et al.

RC2:" Regarding the in-house TC460 standard used for validating the analytical methods. It was clear to me that the variations in B contents in the TC460 shown in Figure 4C were due to supplemental additions of different amounts of the NIST SRM951 reference material. But this appears not the case for the variations in B contents shown in Figure 4A. Then, do they simply reflect the differences in the amounts of TC460 dissolved?

The variations in B content shown in Figure 4A reflect variations in the amount of boron loaded onto the columns. We will make this clear in the revised manuscript.

RC2:"I feel the manuscript will significantly benefit from having a short paragraph describing the process of how diatoms produce their frustules, including how silicic acid is gained from seawater into the cell and what happens afterwards (especially in the

Silicate Deposition Vesicles: "SDV") to deposit silica frustules. Perhaps, the paragraph should also have brief description of carbon acquisition for photosynthesis (Line 364-371), to which an active intake of seawater borate is linked (Line 364-372)."

These are good points and something we can add to the revised manuscript relatively easily.

RC2:"I have several questions and unclear points in the Section 3.2.2. . ... I think it is safe for authors to acknowledge that assumption #1 is an entirely open question at this stage. Furthermore, I wonder how assumption #2 is possible. I presume SDV plays central roles for frustule formation (silica polymerization?). But it is mentioned that the internal pH of SDV should be around 5.5 (Line 334, Line 347). At such pH, essentially all of dissolved B should exist as B(OH)3, not B(OH)4$-$ (Fig. 6). How can you incorporate something that does not exist?"

We readily acknowledge that our model is preliminary and contains assumptions that will be difficult to test. The reviewer however has identified an interesting point here that although we were aware of, for the sake of simplicity we did not expand on in our original model description. At pH 5.5 although the concentration of B(OH)4- is low it is still present, i.e. at S = 35 and T= 20, B(OH)4- = 0.3 umol/kg (around 200x lower concentration than at pH 8). While this perhaps does present a challenge to our model, we should point out that when purifying boron for MC-ICPMS analysis we load our samples at pH 5.5 onto anion exchange columns. Since these use an anion exchange resin with a high partition coefficient for borate, they rapidly remove the small amounts of borate in solution. The remaining dissolved boron respeciates at a fast rate to maintain equilibrium, and this produces more borate that in turn removed by the resin. This cycle repeats until all the boron is removed from the solution, and this occurs at such speed that we have quantitative boron removal in the 10-minute transit time through the resin bed. While we acknowledge the partition coefficients involved in silica deposition in diatoms are unlikely to be as high as for our boron-specific resin, this does at least offer a mechanism for how significant amounts of boron could be

incorporated as borate even though the pH is low in the SDV. We will add more detail in this regard to the revised manuscript with the new schematic discussed in response to RC1.

In response to the other part of this comment – the SDV does play a very important role in the formation of the opal frustule. This will be made clear in the revised manuscript.

The remainder of RC2's comments are minor and will be corrected in the revised manuscript.

---

## Author Response (AR1)

Response to Reviews for "The pH dependency of the boron isotopic composition of diatom opal
(Thalassiosira weissflogii) by Donald et al.

**Reviewer RC1 – Jan Fietzke**

We thank the reviewer for his comments and in particular his recognition that, while this manuscript
is a start of the exploitation of boron isotopes in diatoms, it is not the final word on the matter.  We
respond to each of his comments  in turn below.

RC1: "I had the opportunity to analyze cultured T. weissflogii samples for their boron isotopic
composition using LA-MC-ICPMS about 7 years ago. On average those analyses resulted in d11B of
14.0 +- 1.1 (1sd). Unfortunately, the results appeared too imprecise to be useful and never got
published... Nevertheless, the significant difference of both, own data and those reported in this
manuscript, is quite striking to me."

This is indeed quite a difference, but without more detail it is hard for us to critically evaluate this
observation.  However, we would like to point out that: (i) we carried out an extensive cleaning
protocol to remove residual organic material; (ii) we carried out an extensive investigation of our
protocol including ensuring little to no boron was lost during the purification process; and (iii) our
standard addition tests support the conclusion that our $\delta^{11}B$ analytical method is accurate.  It is also
perhaps worth noting that all published $\delta^{11}B$ measurements to date are also isotopically light, like
our results – though we acknowledge that this is a rather limited dataset to make such comparisons.
In the future we would welcome engaging with the community to further explore the analytical
accuracy of $\delta^{11}B$ in opal-matrices by various analytical techniques.

RC1:"I would like to see in an additional figure a direct comparison of the [B] vs. pH systematic
reported by Meija et al. (2013) and this study."

This was included in the original manuscript as a supplementary figure.  Given this comment (and a
similar one by reviewer RC2) we have now brought this data into Figure 5 (grey circles).

RC1: "The authors suggest the differences may be due to the use of LA and conventional ICPMS. I do
not think, the LA results published by Meija et al. (2013) are inaccurate."

This is not actually what we say in the manuscript, we were careful not to apportion cause and
instead we said the following: "In detail, however, our concentrations are around 2-3 times lower
than Meija et al. (2013), perhaps due to the different analytical methods used (laser ablation ICP-MS
vs. solution here…".

RC1: "So, this would indicate three possibilties: a) samples for the older LA study had not been
cleaned sufficiently (which I doubt strongly) b) the sample preparation used in this study resulted in
a loss of boron or c) some details in the culturing setups resulted in this observable differences."

Since we have not done a direct comparison of methods for determining B/Si it is hard to determine
the specific cause. However, given this comment, we now briefly discuss these possibilities in the
manuscript, expanding on the observed discrepancy (in absolute B/Si but not in the relationship
between B/Si and pH) in lines 289-292.

RC1:"I would also be interested to see a figure displaying d11B vs. [B]. From figure 5 it appears there
may be a stronger correlation of those two parameters than the ones of each of both vs. pH."

There is in fact a weaker relationship between these two variables than between each and pH.  We
therefore decided not to include this figure as it distracted from the good relationships with pH.

RC1: "The model proposed to explain the data (including the -10permill offset during incorporation
into opal) is not really satisfying... This would need a better, more detailed description,maybe
including a schematic figure for a better conceptual understanding."

We have now included a schematic (Figure 8) and have modified the relevant section to improve
clarity (also in accordance with the comments of RC2 below).

**Reviewer RC2 – Joji Uchikawa**

We thank this reviewer for his comments and we are pleased that he favours publication and
recognises the importance of this contribution.  In order to harvest sufficient biomass from our
dilute batch cultures we had to let them continue longer than we would have if we were going to
avoid significant pH drift.  We agree with the reviewer that this is a weakness of this current study
but we do acknowledge this in the manuscript and recommend a different approach for subsequent
studies.  We respond to the rest of RC2's in turn below.

RC2:"I would love to see figure-S1 in the main body of the paper (with proper data legend), which
nicely compares the [B] vs pH relationship revealed here and previously in Mejía et al. (2013). Or
ternatively, I recommend to include the data by Mejía et al. (2013) in Figure 5B"

As mentioned in our response to RC1, we now include these data in Figure 5 of the revised
manuscript.

RC2: "Figure 5: It might be worthwhile to add horizontal error bars (uncertainties for the projected
11B values for borate due to the drifts in pH)."

This is a good idea, these are now added.

RC2: "In addition, I am curious if the authors considered more of a threshold-type response (e.g.,
step-wise increase) for the relationship between diatom B contents and seawater pH (Panel B),
rather than linear regression?"

This is an interesting point but given the uncertainties in both pH and [B] we would prefer not to
overly fit curves to the observed dataset. The statistics suggest a linear fit describes the data
adequately (albeit with some scatter) and such a two stepped relationship is not consistent with the
data from Mejia et al.

RC2:" Regarding the in-house TC460 standard used for validating the analytical methods. It was clear
to me that the variations in B contents in the TC460 shown in Figure 4C were due to supplemental
additions of different amounts of the NIST SRM951 reference material. But this appears not the case
for the variations in B contents shown in Figure 4A. Then, do they simply reflect the differences in
the amounts of TC460 dissolved?

The variations in B content shown in Figure 4A reflect variations in the amount of boron loaded onto
the columns.  We have attempted to make this clearer in the figure caption of this figure (line 471).

RC2:"I feel the manuscript will significantly benefit from having a short paragraph describing the
process of how diatoms produce their frustules, including how silicic acid is gained from seawater
into the cell and what happens afterwards (especially in the Silicate Deposition Vesicles: "SDV") to
deposit silica frustules. Perhaps, the paragraph should also have brief description of carbon
acquisition for photosynthesis (Line 364-371), to which an active intake of seawater borate is linked
(Line 364-372)."

These are good points and we have now added this 338-343 and 387-388.

RC2:"I have several questions and unclear points in the Section 3.2.2…. I think it is safe for authors to
acknowledge that assumption #1 is an entirely open question at this stage. Furthermore, I wonder
how assumption #2 is possible. I presume SDV plays central roles for frustule formation (silica
polymerization?). But it is mentioned that the internal pH of SDV should be around 5.5 (Line 334,
Line 347). At such pH, essentially all of dissolved B should exist as $B(OH)_3$, not $B(OH)_4^-$ (Fig. 6). How
can you incorporate something that does not exist?"

We now expand on the points raised here in the revised manuscript (lines 377-386).

In response to the other part of this comment – the SDV does play a very important role in the
formation of the opal frustule.  This will be made clear in the revised manuscript.

The remainder of RC2's comments are minor and will be corrected in the revised manuscript.

[revised manuscript text omitted]